# HOI Analysis: Integrating and Decomposing Human-Object Interaction

**Yong-Lu Li**[*]    **Xinpeng Liu**[*]    **Xiaoqian Wu**    **Yizhuo Li**    **Cewu Lu**[†]

Shanghai Jiao Tong University

yonglu_li@sjtu.edu.cn, xinpengliu0907@gmail.com, enlighten@sjtu.edu.cn
liyizhuo@sjtu.edu.cn, lucewu@sjtu.edu.cn

## Abstract

Human-Object Interaction (HOI) consists of human, object and *implicit* interaction/verb. Different from previous methods that directly map pixels to HOI semantics, we propose a novel perspective for HOI learning in an analytical manner. In analogy to Harmonic Analysis, whose goal is to study how to represent the signals with the superposition of basic waves, we propose the **HOI Analysis**. We argue that coherent HOI can be decomposed into isolated human and object. Meanwhile, isolated human and object can also be integrated into coherent HOI again. Moreover, transformations between human-object pairs with the same HOI can also be easier approached with integration and decomposition. As a result, the implicit verb will be represented in the transformation function space. In light of this, we propose an **Integration-Decomposition Network (IDN)** to implement the above transformations and achieve state-of-the-art performance on widely-used HOI detection benchmarks. Code is available at `https://github.com/DirtyHarryLYL/HAKE-Action-Torch/tree/IDN-(Integrating-Decomposing-Network)`.

## 1 Introduction

Human-Object Interaction (HOI) takes up most of the human activities. As a composition, HOI consists of three parts: <human, verb, object>. To detect HOI, machines need to simultaneously locate human and object and classify the verb [4]. Except for the direct thinking that maps pixels to semantics, in this work we rethink HOI and explore two questions in a novel perspective (Fig. 1): First, as for the inner structure of HOI, how do *isolated* human and object compose HOI? Second, what is the relationship between two human-object pairs with the same HOI?

For the **first question**, we may find some clues from psychology. The view of Gestalt psychology is usually summarized as one simple sentence: "*The whole is more than the sum of its parts*" [14]. This is also in line with human perception. Baldassano $et\ al.$ [1] studied the mechanism of how the brain builds HOI representation and concluded that the encoding of HOI is not the simple sum of human and object: a higher-level neural representation exists. Specific brain regions, $e.g.$, posterior superior temporal sulcus (pSTS), are responsible for integrating *isolated* human and object into *coherent* HOI [1]. Hence, to encode HOI, we may need complex nonlinear transformation (**integration**) to combine isolated human and object. Also, we argue that a reverse process is also essential to decompose HOI into isolated human and object (**decomposition**). Here we use $T_I(\cdot)$ and $T_D(\cdot)$ to indicate integration and decomposition functions. According to [1], isolated human and object are different from coherent HOI pair. Therefore, $T_I(\cdot)$ should be able to **add** interactive relationship to isolated elements. On the contrary, $T_D(\cdot)$ should **eliminate** this interactive information. Through the

---

[*]The first two authors contribute equally.

[†]Cewu Lu is the corresponding author, member of Qing Yuan Research Institute and MoE Key Lab of Artificial Intelligence, AI Institute, Shanghai Jiao Tong University, China and Shanghai Qi Zhi institute.

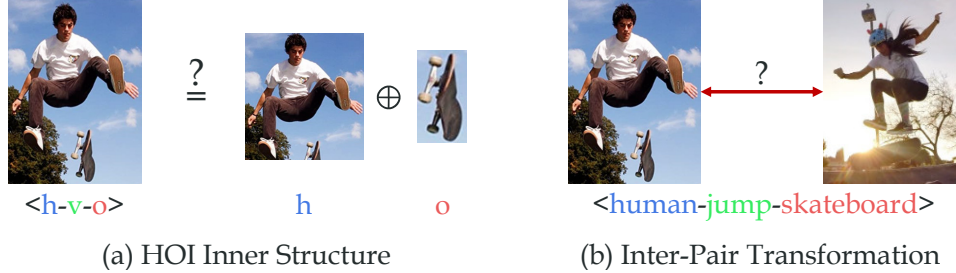

(a) HOI Inner Structure        (b) Inter-Pair Transformation

Figure 1: Two questions about HOI. First, we want to explore its inner structure. Second, the relationship between two human-object pairs with same HOI is studied.

*semantic change* before and after transformations, we can reveal the "eigen" structure of HOI carrying the semantics. Considering that *verb* is hard to represent explicitly in image space, our transformations are conducted in latent space. For the **second question**, directly transforming one human-object pair to another (**inter-pair transformation**) is difficult. We need to consider not only isolated element differences but also interaction pattern change. However, with $T_I(\cdot)$ and $T_D(\cdot)$, things are different. We can first decompose HOI pair-$i$ into isolated person-$i$ and object-$i$ and eliminate the interaction semantics. Next, we transform human-$i$ (object-$i$) to human-$j$ (object-$j$) ($j \neq i$). The last step is to integrate human-$j$ and object-$j$ into pair-$j$ and add the interaction simultaneously.

Interestingly, we find the above process is kind of like *Harmonic Analysis*: to process the signal, we usually use Fourier Transform (FT) to decompose it into the integration of basic exponential functions; then we can modulate the exponential functions via very simple transformations like scalar-multiplication; finally, inverse FT can help us integrate the modulated elements and map them back to the input space. This elegant property brings a lot of convenience for signal processing. Therefore, we mimic this insight and design our methodology, *i.e.*, **HOI Analysis**. To implement HOI Analysis, we propose an **Integration-Decomposition Network** (**IDN**). In detail, after extracting the features from human/object and human-object tight union boxes, we perform the integration $T_I(\cdot)$ to integrate the isolated human and object into the union in latent space. Moreover, decomposition $T_D(\cdot)$ is then performed to decompose the union into isolated human and object instances again. Through the transformations, IDN can learn to represent the interaction/verb with $T_I(\cdot)$ and $T_D(\cdot)$. That said, we first embed verbs in transformation function space, then learn to *add* and *eliminate* interaction semantics and classify interactions during transformations. For the inter-pair transformation, we adopt a simple **instance exchange policy**. For each human/object, we beforehand find its similar instances as candidates and randomly exchange the original instance with candidates in training. This policy can avoid complex transformation like motion transfer [3]. Hence, we can focus on the learning of $T_I(\cdot)$ and $T_D(\cdot)$. Moreover, the lack of samples for rare HOIs can also be alleviated. To train IDN, we adopt the objectives derived from *transformation principles*, such as **integration validity**, **decomposition validity** and **interactiveness validity** (detailed in Sec. 3.4). With them, IDN can effectively model the interaction/verb in transformation function space. Subsequently, IDN can be applied to the HOI detection task by comparing the above validities and greatly advance it.

Our contributions are threefold: (1) Inspired by Harmonic Analysis, we thereon devise HOI Analysis to model the HOI inner structure. (2) A concise Integration-Decomposition Network (IDN) is proposed to conduct the transformations in HOI Analysis. (3) By learning verb representation in transformation function space, IDN achieves state-of-the-art performance on HOI detection.

## 2 Related Work

Human-Object Interaction (HOI) detection [4, 18] is crucial for deeper scene understanding and can facilitate behavior and activity learning [15, 25, 46, 47, 37, 38, 44]. Recently, huge progress has been made in this field with the promotion of large-scale datasets [18, 4, 5, 15, 25] and deep learning. HOI has been studied for a long history. Previously, most methods [16, 17, 55, 54, 6, 7] adopted hand-crafted features. With the renaissance of neural networks, recent works [8, 32, 27, 12, 45, 19, 49, 42, 4, 13, 41, 24, 28] start to leverage learning-based features with end-to-end paradigm. HO-RCNN [4] utilized a multi-stream model to leverage human, object and spatial patterns respectively, which is widely followed by subsequent works [12, 27, 49]. Differently, GPNN [42] adopted a graph model to address HOI learning for both images and videos. Instead of directly processing all

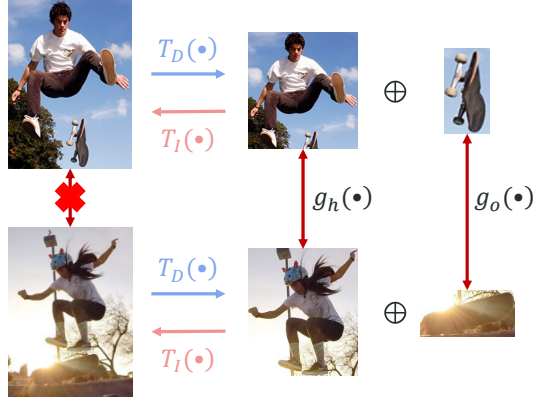

Figure 2: HOI Analysis. $T_D(\cdot)$ and $T_I(\cdot)$ indicate the decomposition and integration. First, we decompose the coherent HOI into isolated human and object. Next, human and object can be integrated into HOI again. Through $T_D(\cdot)$ and $T_I(\cdot)$, we can model the verb in transformation function space and conduct the inter-pair transformation (IPT) more easily. Red "X" means it is hard to operate IPT directly. $g_h$, $g_o$ indicate the inter-human/object transformation functions.

human-object pairs generated from detection, TIN [27] utilized interactiveness estimation to filter out non-interactive pairs in advance. In terms of modality, Peyre *et al.* [41] explored to learn a joint space via aligning the visual and linguistic features and used word analogy to address unseen HOIs. DJ-RN [24] recovered 3D human and object (location and size) and learned a 2D-3D joint representation. Finally, some works also explore to encode HOI with the help of a knowledge base. Based on human part-level semantics, HAKE [25] built a large-scale part state [30] knowledge base and Activity2Vec for finer-grained action encoding. Xu *et al.* [53] constructed a knowledge graph from HOI annotations and the external source to advance the learning.

Besides the computer vision community, HOI is also studied in human perception and cognition researches. In [1], Baldassano *et al.* studied how human brain models HOI given HOI images. Interestingly, besides the brain regions responsible for encoding isolated human or object, certain regions can integrate isolated human and object into a higher-level joint representation. For example, pSTS can coherently model the HOI, instead of simply summing isolated human and object information. This phenomenon inspires us to rethink the nature of HOI representation. Thus we propose a novel HOI Analysis method to encode HOI by integration and decomposition.

On the other hand, HOI learning is similar to another compositional problem: attribute-object learning [35, 34, 26]. Attribute-object compositions have many interesting properties such as contextuality, compositionality [34, 35] and symmetry [26]. To learn the attribute-object, attributes are seen as primitives equal with objects [34] or linear/non-linear transformations [35, 26]. Different from attributes expressed on object appearance, verbs in HOIs are more implicit and hard to locate in images. They are a kind of *holistic* representation of composed human and object instances. Thus, we propose several transformation validities to embed and capture the verbs in transformation function space, instead of utilizing an explicit classifier to classify them [34] or using language priors [35, 26].

## 3 Method

### 3.1 Overview

In an image, human and object can be explicitly seen. However, we can hardly depict which region is the verb. For "hold cup", "hold" may be obvious and center on the hand and cup. But for "ride bicycle", most parts of the person and bicycle all represent "ride". Hence, vision systems may struggle given diverse interactions as it is hard to capture the appropriate visual regions. Though attention mechanism [12] may help, the long-tail distribution of HOI data usually makes it unstable. In this work, instead of directly finding the interaction region and mapping it to semantics [4, 12, 27, 49, 19], we propose a novel learning paradigm, *i.e.*, learning the verb representation via **HOI Analysis**.

Inspired by the perception study [1], we propose the integration $T_I(\cdot)$ and decomposition $T_D(\cdot)$ functions. HOI naturally consists of human, object and implicit verb. Thus, we can decompose HOI

into basic elements and integrate them again like Harmonic Analysis. The overview of HOI Analysis is depicted in Fig. 2. As HOI is not the simple sum of isolated human and object [1], different from FT, our transformations are **nonequivalent**. The *key difference* lies in the *addition* and *elimination* of implicit interactions. We use binary *interactiveness* [27], which indicates whether human and object are interactive, to monitor these semantic changes. Hence, the interactiveness [27] of isolated human/object is *False*, and joint human-object has *True* interactiveness. From the above, $T_I(\cdot)$ should have the ability to "add" interaction to isolated instances and make the integrated human-object has *True* interactiveness. On the contrary, $T_D(\cdot)$ can "eliminate" the interaction between coherent human-object and force their interactiveness to be *False*. At last, to encode the implicit verbs, we represent them in the **transformation function space**. A *pair* of decomposition and integration functions are constructed for *each verb* and forced to operate the appropriate transformations.

We introduce the feature preparation as follows. **First**, given an image, we use an object detector [43] to obtain the human/object boxes $b_h, b_o$. Then, we adopt a COCO [29] pre-trained ResNet-50 [20] to extract human/object RoI pooling features $f_h^a, f_o^a$ from the *third* ResNet Block, where $a$ indicates visual appearance. For simplicity, we use *tight union box* of human and object to represent the coherent HOI pair (**union**). Notably, *coherent* HOI carries the interaction semantics and is **more** than the sum of isolated human and object [1], *i.e.*, the *incoherent* ones. With $b_h, b_o$, the union box $b_u$ can be easily obtained. The RoI pooling feature of $b_u$ is thus adopted from the *fourth* ResNet Block as the appearance representation of coherent HOI ($f_u^a$). Note that $f_u^a$ is twice the size of $f_h^a, f_o^a$, for passing through one more ResNet Block. **Second**, to encode the box location, we generate location features $f_h^b, f_o^b, f_u^b$, where $b$ indicates box location. We follow the box coordinate normalization method [41], getting the normalized box $\hat{b_h}, \hat{b_o}$. Next, for union box, we concatenate $\hat{b_h}$ and $\hat{b_o}$ and feed them to an MLP to get $f_u^b$. For human/object box, $\hat{b_h}$ or $\hat{b_o}$ is also fed to an MLP to get $f_h^b$ or $f_o^b$. The size of $f_h^b$ or $f_o^b$ is half the size of $f_u^b$. **Third**, the location features $f_u^b, f_h^b, f_o^b$ are concatenated respectively to their corresponding appearance features $f_u^a, f_h^a, f_o^a$, getting $\hat{f_u}, \hat{f_h}, \hat{f_o}$. The size of $\hat{f_h}$ and $\hat{f_o}$ are also half the size of $\hat{f_u}$. For convenience, we concatenate $\hat{f_h}$ and $\hat{f_o}$ as $\hat{f_h} \oplus \hat{f_o}$.

Before transformations, we compress these features to reduce the computational burden via an auto-encoder (AE). This AE is given $\hat{f_u}$ as input and pre-trained with an input-output reconstruction loss and a verb classification loss (Sec. 3.4). The classification score is denoted as $S_v^{AE}$. After pre-training, we use AE to compress $\hat{f_u}$ and $\hat{f_h} \oplus \hat{f_o}$ to 1024 sized $f_u$ (coherent) and $f_h \oplus f_o$ (isolated) respectively, Finally, we have $f_u, f_h \oplus f_o$ for integration and decomposition. The **ideal** transformations are:

$$T_D(f_u) = f_h \oplus f_o, T_I(f_h \oplus f_o) = f_u, \tag{1}$$

where $T_D(\cdot), T_I(\cdot)$ indicates the decomposition and integration functions, $\oplus$ indicates the linear operation between isolated human and object features such as element-wise summation or concatenation. In most cases, concatenation performs better. As for the inter-pair transformation, we use

$$g_h(f_h^i) = f_h^j, g_o(f_o^i) = f_o^j, i \neq j, \tag{2}$$

where $f_h^i, f_o^i$ indicate the features of human/object instances, and $g_h(\cdot), g_o(\cdot)$ are the inter-human/object transformation functions. Because the *strict* inter-pair transformation like motion transfer [3] is complex and not our main goal, we implement $g_h(\cdot)$ and $g_o(\cdot)$ as simple *feature replacement* for simplicity. For human instances, we find their substitutional persons with the same HOI according to the pose similarity. As to object instances, we use the objects of the same category and similar sizes as the substitutions. All substitutional candidates come from the **same dataset** (train set) and are randomly sampled during training. From the experiment (Sec. 4.5), we find that this policy performs well and effectively improves the interaction representation learning.

We propose a concise Integration-Decomposition Network (IDN) as shown in Fig. 3. IDN mainly consists of two parts: the first one is the integration and decomposition transformations (Sec. 3.2) which construct a loop between the union and human/object features; the second one is the inter-pair transformation (Sec. 3.3) that exchanges the human/object instances between pairs with same HOI. In Sec 3.4, we introduce the training objectives derived from the transformation principles. With them, IDN would learn more effective interaction representations and advance HOI detection in Sec. 3.5.

## 3.2 Integration and Decomposition

As shown in Fig. 3, IDN constructs a loop consists of two inverse transformations: integration and decomposition implemented with MLPs. That is, we represent the verb/interaction in MLP *weight*

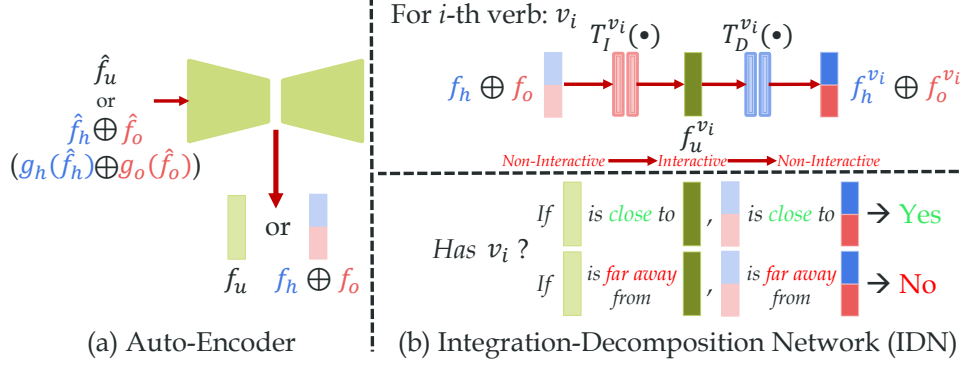

(a) Auto-Encoder | (b) Integration-Decomposition Network (IDN)

Figure 3: The structure of IDN. (a) depicts the feature compressor AE. (b) shows the integration-decomposition loop. For each verb $v_i$, we adopt corresponding $T_I^{v_i}(\cdot)$ and $T_D^{v_i}(\cdot)$. The L2 distance $d_u^{v_i}, d_{ho}^{v_i}$ are then used in interaction classification (Sec. 3.5). Notably, the encoded feature $f_h \oplus f_o$ is the **sum** of isolated human and object and thus not yet integrated with the HOI semantics (Fig. 4).

*space* or *transformation function space*. For **each** verb, we adopt **a pair** of appropriative MLPs as integration and decomposition functions, *e.g.*, $T_I^{v_i}(\cdot)$ and $T_D^{v_i}(\cdot)$ for verb $v_i$. For integration, when inputting a pair of isolated $f_h$ and $f_o$, $\{T_I^{v_i}(\cdot)\}_{i=1}^n$ integrates them into $n$ outputs for $n$ verbs:

$$f_u^{v_i} = T_I^{v_i}(f_h \oplus f_o), \tag{3}$$

where $i = 1, 2, 3, ....n$ and $n$ is the number of verbs, $f_u^{v_i}$ is the *integrated* union feature for the $i$-$th$ verb. $\oplus$ indicates concatenation. Through the integration function set $\{T_I^{v_i}(\cdot)\}_{i=1}^n$, we get a set of 1024 sized integrated union features $\{f_u^{v_i}\}_{i=1}^n$. If the original $f_u$ contains the semantics of the $i$-$th$ verb, it should be **close** to $f_u^{v_i}$ and **far** away from the other integrated union features. Second, the subsequent decomposition is depicted as follows. Given the integrated union feature set $\{f_u^{v_i}\}_{i=1}^n$, we also use $n$ decomposition functions $\{T_D^{v_i}(\cdot)\}_{i=1}^n$ to decompose them respectively:

$$f_h^{v_i} \oplus f_o^{v_i} = T_D^{v_i}(f_u^{v_i}). \tag{4}$$

The decomposition output is also a set of features $\{f_h^{v_i} \oplus f_o^{v_i}\}_{i=1}^n$, where $f_h^{v_i}$ and $f_o^{v_i}$ all have the same size with $f_h$ and $f_o$. Similarly, if this human-object pair is performing the $i$-$th$ interaction, the original input $f_h \oplus f_o$ should be **close** to the $f_h^{v_i} \oplus f_o^{v_i}$ and **far** away from the other $\{f_h^{v_j} \oplus f_o^{v_j}\}_{j \neq i}$.

### 3.3 Inter-Pair Transformation

Inter-Pair Transformation (IPT) is proposed to reveal the inherent nature of implicit verb, *i.e.*, the **shared information** between different pairs with the same HOI. Here, we adopt a simple implementation: *instance exchange policy*. For humans, we first use pose estimation [9, 23] to obtain poses and then operate alignment and normalization. In detail, the pelvis keypoints of all persons are aligned and all the distances between head and pelvis are scaled to one. Hence, we can find similar persons according to the pose similarity, which is calculated as the sum of Euclidean distances between the corresponding keypoints of two persons. To keep the semantics, similar persons should have at least one same HOI. Selecting similar objects is simpler, we directly choose the objects of the same category. An extra criterion is that we choose objects with similar sizes. We use the area ratio between the object box and the paired human box as the criteria. Finally, $m$ similar candidates are selected for each human/object. This whole selection is operated within *one dataset*. Formally, with instance exchange, Eq. 3 can be rewritten as:

$$f_u^{v_i} = T_I^{v_i}(g_h(f_h) \oplus g_o(f_o)) = T_I^{v_i}(f_h^{k_1} \oplus f_o^{k_2}), \tag{5}$$

where $k_1, k_2 = 1, 2, 3, ...m$ and $m$ is the number of selected similar candidates, here $m = 5$. And $\{T_I^{v_i}(\cdot)\}_{i=1}^n$ and $\{T_D^{v_i}(\cdot)\}_{i=1}^n$ should be **equally effective** before and after instance exchange. During training, we first use Eq. 3 for a certain number of epochs and then replace Eq. 3 with Eq. 5 (Sec. 4.2). When using Eq. 5, we put the original instance and its exchanging candidates together and randomly sample them. Notably, we focus on the transformations between pairs with the *same* HOI. The transformations between *different* HOIs which need to manipulate the corresponding human posture, human-object spatial configuration and interactive pattern are beyond the scope of this paper. For IPT, more sophisticate approaches are also possible, *e.g.*, using motion transfer [3] to adjust

2D human posture according to another person with the same HOI but different posture (eating while sitting/standing), recovering 3D HOI [39, 24] and adjusting 3D pose [22, 10] to generate new images/features, using language priors to change the classes of interacted objects or HOI compositions [2, 36], *etc*. But these are beyond the scope of our main insight, so we leave these to the future work.

## 3.4 Transformation Principles as Objectives

Before training, we first pre-train AE to compress the inputs. We first feed $\hat{f}_u$ to the encoder and obtain the compressed $f_u$. Then an MLP takes $f_u$ as input to classify the verbs with Sigmoids (one pair can have multiple HOIs simultaneously) with cross-entropy loss $L_{cls}^{AE}$. Meanwhile, $f_u$ is decoded and generates $f_u^{recon}$. We construct MSE reconstruction loss $L_{recon}^{AE}$ between $f_u^{recon}$ and $\hat{f}_u$. The overall loss of AE is $L^{AE} = L_{cls}^{AE} + L_{recon}^{AE}$. After pre-training, AE will be fine-tuned together with transformation modules. Next, we detail the objectives derived from transformation principles.

**Integration Validity.** As aforementioned, we integrate $f_h$ and $f_o$ into the union feature set $\{f_u^{v_i}\}_{i=1}^n$ for all verbs (Eq. 3 or 5). If integration is able to "add" the verb semantics, the corresponding $f_u^{v_i}$ that belongs to the *ongoing* verb classes should be close to the real $f_u$. For example, if coherent $f_u$ contains the semantics of verb $v_p$ and $v_q$, then $f_u^{v_p}$ and $f_u^{v_q}$ should be close to $f_u$. Meanwhile, $\{f_u^{v_i}\}_{i\neq p,q}^n$ should be far away from $f_u$. Hence, we can construct the distance:

$$d_u^{v_i} = ||f_u - f_u^{v_i}||_2. \tag{6}$$

For $n$ verb classes, we can get distance set $\{d_u^{v_i}\}_{i=1}^n$. Considering above principle, if $f_u$ carries the $p$-$th$ verb semantics, $d_u^p$ should be small, and vice versa. Therefore, we can directly use the **negative** distances as the score of verb classification, *i.e.* $S_v^u = \{-d_u^{v_i}\}_{i=1}^n$. Naturally, $S_v^u$ is then used to generate verb classification loss $L_{cls}^u = L_{ent}^u + L_{hinge}^u$, where $L_{ent}^u$ is cross-entropy loss, $L_{hinge}^u = \sum_{i=1}^n[y_i \max(0, d^{v_i} - t_1^{v_i}) + (1 - y_i)\max(0, t_0^{v_i} - d^{v_i})]$. $y_i = 1$ indicates this pair has verb $v_i$ and otherwise $y_i = 0$. $t_0$ and $t_1$ are chosen following semi-hard mining strategy: $t_1^{v_i} = \min_{B_-^{v_i}}(d^{v_i})$ and $t_0^{v_i} = \max_{B_+^{v_i}}(d^{v_i})$, where $B_-^{v_i}$ denotes all the pairs without verb $v_i$ in the current mini-batch, and $B_+^{v_i}$ denotes all the pairs with verb $v_i$ in the current mini-batch.

**Decomposition Validity.** This validity is proposed to constrain the decomposed $\{f_h^{v_i} \oplus f_o^{v_i}\}_{i=1}^n$ (Eq. 4). Similar to Eq. 6, we also construct $n$ distances between $\{f_h^{v_i} \oplus f_o^{v_i}\}_{i=1}^n$ and $f_h \oplus f_o$ as

$$d_{ho}^{v_i} = ||f_h \oplus f_o - f_h^{v_i} \oplus f_o^{v_i}||_2 \tag{7}$$

and obtain $\{d_{ho}^{v_i}\}_{i=1}^n$. Again $\{d_{ho}^{v_i}\}_{i=1}^n$ should obey the same principle according to ongoing verbs. Thus, we get the second verb score $S_v^{ho} = \{-d_{ho}^{v_i}\}_{i=1}^n$ and verb classification loss $L_{cls}^{ho}$.

**Interactiveness Validity.** Interactiveness [27] depicts whether a person and an object are interactive. Thus, it is *False* if and only if human-object do not have any interactions. As the "1+1>2" property [1], the interactiveness of isolated human $f_h$ or object $f_o$ should be *False*, so does $f_h \oplus f_o$. But after we integrate $f_h \oplus f_o$ into $\{f_u^{v_i}\}_{i=1}^n$, its interactiveness should be *True*. Meanwhile, the original union $f_u$ should have *True* interactiveness. We adopt one *shared* FC-Sigmoid as the binary classifier for $f_u$, $f_h \oplus f_o$ and $\{f_u^{v_i}\}_{i=1}^n$. The binary label converted from HOI label is *zero* if and only if a pair does not have any interactions. Notably, we also adopt the interactiveness validity upon decomposed $\{f_h^{v_i} \oplus f_o^{v_i}\}_{i=1}^n$ but achieve limited improvement. To keep the model concise, we just adopt the other three effective interactiveness validities hereinafter. Thus, we obtain three binary classification cross entropy losses: $L_{bin}^u, L_{bin}^{ho}, L_{bin}^I$. For clarity, we use a unified $L_{bin} = L_{bin}^u + L_{bin}^{ho} + L_{bin}^I$.

The overall loss of IDN is $L = L_{cls}^u + L_{cls}^{ho} + L_{bin}$. With the guidance of these principles, IDN can well capture the *interaction changes* during the transformations. Different from previous methods that aim at encoding the entire HOI representations *statically*, IDN focuses on *dynamically* inferring whether an interaction exists within human-object through the integration and decomposition. So IDN can alleviate the learning difficulty of complex and various HOI patterns.

## 3.5 Application: HOI Detection

We further apply IDN to HOI detection, which needs to simultaneously locate human-object and classify the ongoing interactions. For locations, we adopt the detected boxes from a COCO [29]

pre-trained Faster R-CNN [43], so does the object class probability $P_o$. Then, verb scores can be obtained from Eq. 6 and 7. $S_v^u = \{-d_u^{v_i}\}_{i=1}^n$, $S_v^{ho} = \{-d_{ho}^{v_i}\}_{i=1}^n$ and $S_v^{AE}$ obtained from AE are then fed to exponential functions or Sigmoids to generate $P_v^u = \exp(S_v^u)$, $P_v^{ho} = \exp(S_v^{ho})$ and $P_v^{AE} = Sigmoid(S_v^{AE})$. Since the validity losses would *pull* the features that meet the labels together and *push* away the others, thus here we directly use three kinds of distances to classify verbs. For example, if $f_u$ contains the $i$-th verb, $d_u^{v_i} = ||f_u - f_u^{v_i}||_2$ should be *small* (probability should be **large**); if not, $d_u^{v_i}$ should be *large* (probability should be **small**). The final verb probabilities is acquired via $P_v = \alpha(P_v^u + P_v^{ho} + P_v^{AE})$, here $\alpha = \frac{1}{3}$. For HOI triplets, we get their HOI probabilities using $P_{HOI} = P_v * P_o$ for all *possible* compositions according to the benchmark setting.

# 4 Experiment

In this section, we first introduce the adopted datasets, metrics (Sec. 4.1) and implementation (Sec. 4.2). Next, we compare IDN with the state-of-the-art on HICO-DET [4] and V-COCO [18] in Sec. 4.3. As HOI detection metrics [4, 18] expect both accurate human/object locations and verb classification, the performance strongly relies on object detection. Hence, we conduct experiments to evaluate IDN with different object detectors. At last, ablation studies are conducted (Sec. 4.5).

## 4.1 Dataset and Metric

We adopt the widely-used HICO-DET [4] and V-COCO [18]. **HICO-DET [4]** consists of 47,776 images (38,118 for training and 9,658 for testing) and 600 HOI categories (80 COCO [29] objects and 117 verbs). **V-COCO [18]** contains 10,346 images (2,533 and 2,867 in train and validation sets, 4,946 in test set). Its annotations include 29 verb categories (25 HOIs and 4 body motions) and same 80 objects with HICO-DET[4]. For HICO-DET, we use mAP following [4]: true positive needs to contain accurate human and object locations (box IoU with reference to GT box is larger than 0.5) and accurate verb classification. The role means average precision [18] is used for V-COCO.

## 4.2 Implementation Details

The encoder of the adopted AE compresses the input feature dimension from 4608 to 4096, then to 1024. The decoder is structured symmetrical to the encoder. For HICO-DET [4], AE is pre-trained for 4 epochs using SGD with a learning rate of 0.1, momentum of 0.9, while each batch contains 45 positive and 360 negative pairs. The whole IDN (AE and transformation modules) is first trained without inter-pair transformation (IPT) for 20 epochs using SGD with a learning rate of 2e-2, momentum of 0.9. Then we finetune IDN with IPT for 30 epochs using SGD, with a learning rate of 1e-3, momentum of 0.9. Each batch for the whole IDN contains 15 positive and 120 negative pairs. For V-COCO [18], AE is first pre-trained for 60 epochs. The whole IDN is trained without IPT for 45 epochs using SGD, then fine-tuned with IPT for 20 epochs. The other training parameters are the same as those for HICO-DET. In testing, LIS [27] is adopted with $T = 8.3, k=12.0, \omega=10.0$. Following [27], we use NIS [27] in all testings with the default threshold and the interactiveness estimation of $f_u$. All experiments are conducted on one single NVIDIA Titan Xp GPU.

## 4.3 Results

**Setting.** We compare IDN with state-of-the-art [45, 4, 13, 42, 12, 27, 19, 49, 41, 48, 24, 11, 21, 53, 50, 56, 28, 51, 18, 49] on two benchmarks in Tab. 1 and Tab. 2. For HICO-DET, we follow the settings in [4]: Full (600 HOIs), Rare (138 HOIs), Non-Rare (462 HOIs) in Default and Known Object sets. For V-COCO, we evaluate $AP_{role}$ (24 actions with roles) on Scenario 1 (S1) and Scenario 2 (S2). To purely illustrate the HOI recognition ability without the influence of object detection, we conduct evaluations with *three kinds of detectors*: COCO pre-trained (**COCO**), pre-trained on COCO and then *finetuned* on HICO-DET train set (**HICO-DET**), GT boxes (**GT**) in Tab. 1.

**Comparison.** With $T_I(\cdot)$ and $T_D(\cdot)$, IDN outperforms previous methods significantly and achieves **23.36** mAP on the Default Full set of HICO-DET [4] with COCO detector. Moreover, IDN is the first to achieve more than 20 mAP on all three Default sets without additional information used. Moreover, the improvement on the Rare set proves that the *dynamically* learned interaction representation can greatly alleviate the data deficiency of rare HOIs. With the HICO-DET finetuned detector, IDN also shows great improvements and achieves more than **26** mAP and further proves

| Method | Detector | Feature | Default Full | | | Known Object | | |
|---|---|---|---|---|---|---|---|---|
| | | | Full | Rare | Non-Rare | Full | Rare | Non-Rare |
| Shen *et al.* [45] | COCO | VGG-19 | 6.46 | 4.24 | 7.12 | - | - | - |
| HO-RCNN [4] | COCO | CaffeNet | 7.81 | 5.37 | 8.54 | 10.41 | 8.94 | 10.85 |
| InteractNet [13] | COCO | ResNet50-FPN | 9.94 | 7.16 | 10.77 | - | - | - |
| GPNN [42] | COCO | ResNet101 | 13.11 | 9.34 | 14.23 | - | - | - |
| Xu *et al.* [53] | COCO | ResNet50 | 14.70 | 13.26 | 15.13 | - | - | - |
| iCAN [12] | COCO | ResNet50 | 14.84 | 10.45 | 16.15 | 16.26 | 11.33 | 17.73 |
| Wang *et al.* [50] | COCO | ResNet50 | 16.24 | 11.16 | 17.75 | 17.73 | 12.78 | 19.21 |
| TIN [27] | COCO | ResNet50 | 17.03 | 13.42 | 18.11 | 19.17 | 15.51 | 20.26 |
| No-Frills [19] | COCO | ResNet152 | 17.18 | 12.17 | 18.68 | - | - | - |
| Zhou *et al.* [56] | COCO | ResNet50 | 17.35 | 12.78 | 18.71 | - | - | - |
| PMFNet [49] | COCO | ResNet50-FPN | 17.46 | 15.65 | 18.00 | 20.34 | 17.47 | 21.20 |
| DRG [11] | COCO | ResNet50-FPN | 19.26 | 17.74 | 19.71 | 23.40 | 21.75 | 23.89 |
| Peyre *et al.* [41] | COCO | ResNet50-FPN | 19.40 | 14.60 | 20.90 | - | - | - |
| VCL [21] | COCO | ResNet50 | 19.43 | 16.55 | 20.29 | 22.00 | 19.09 | 22.87 |
| VSGNet [48] | COCO | ResNet152 | 19.80 | 16.05 | 20.91 | - | - | - |
| DJ-RN [24] | COCO | ResNet50 | 21.34 | 18.53 | 22.18 | 23.69 | 20.64 | 24.60 |
| **IDN** | COCO | ResNet50 | **23.36** | **22.47** | **23.63** | **26.43** | **25.01** | **26.85** |
| PPDM [28] | HICO-DET | Hourglass-104 | 21.73 | 13.78 | 24.10 | 24.58 | 16.65 | 26.84 |
| Bansal *et al.* [2] | HICO-DET | ResNet101 | 21.96 | 16.43 | 23.62 | - | - | - |
| TIN [27]$^{\text{VCL}}$ | HICO-DET | ResNet50 | 22.90 | 14.97 | 25.26 | 25.63 | 17.87 | 28.01 |
| TIN [27]$^{\text{DRG}}$ | HICO-DET | ResNet50 | 23.17 | 15.02 | 25.61 | 24.76 | 16.01 | 27.37 |
| VCL [21] | HICO-DET | ResNet50 | 23.63 | 17.21 | 25.55 | 25.98 | 19.12 | 28.03 |
| DRG [11] | HICO-DET | ResNet50-FPN | 24.53 | 19.47 | 26.04 | 27.98 | 23.11 | **29.43** |
| **IDN**$^{\text{VCL}}$ | HICO-DET | ResNet50 | 24.58 | 20.33 | 25.86 | 27.89 | 23.64 | 29.16 |
| **IDN**$^{\text{DRG}}$ | HICO-DET | ResNet50 | **26.29** | **22.61** | **27.39** | **28.24** | **24.47** | 29.37 |
| iCAN [12] | GT | ResNet50 | 33.38 | 21.43 | 36.95 | - | - | - |
| TIN [27] | GT | ResNet50 | 34.26 | 22.90 | 37.65 | - | - | - |
| Peyre *et al.* [41] | GT | ResNet50-FPN | 34.35 | 27.57 | 36.38 | - | - | - |
| **IDN** | GT | ResNet50 | **43.98** | **40.27** | **45.09** | - | - | - |

Table 1: Results on HICO-DET [4]. "COCO" is the COCO pre-trained detector, "HICO-DET" means that the "COCO" is further fine-tuned on HICO-DET, "GT" means the ground truth human-object box pairs. Superscript $^{\text{DRG}}$ or $^{\text{VCL}}$ indicates that the HICO-DET fine-tuned detector from DRG [11] or VCL [21] is used.

the affect from detections (23.36 to 26.29 mAP). Given GT boxes, the gaps among the other three methods [12, 27, 41] are marginal. But IDN achieves more than **9** mAP improvement on HOI recognition solely. All these greatly verify the efficacy of our integration and decomposition. On V-COCO [18], IDN achieves **53.3** mAP on S1 and **60.3** mAP on S2, both significantly outperforming previous methods. Moreover, we also apply our IDN to existing HOI methods since its flexibility as a plug-in. In detail, we apply integration and decomposition to iCAN [12] as a **proxy task** to enhance its feature learning. The performance improves from 14.84 mAP to **18.98** mAP (HICO-DET Full).

**Efficiency and Scalability.** In IDN, each verb is represented by a pair of MLPs ($T_I^{v_i}(\cdot)$ and $T_D^{v_i}(\cdot)$). To ensure the efficiency, we carefully designed the data flow to make IDN is able to run on a **single** GPU. All transformations are operated **in parallel** and the inference speed is **10.04** FPS (iCAN [12]: 4.90 FPS, TIN [27]: 1.95 FPS, PPDM [28]: 14.08 FPS, PMFNet [49]: 3.95 FPS). We also considered an implementation which utilizes **a single MLP for all verbs** for scalability, *i.e.*, conditioned MLP functions $f_u^{v_i} = T_I(f_h \oplus f_o, f_{v_i}^{ID})$ and $f_h^{v_i} \oplus f_o^{v_i} = T_D(f_u^{v_i}, f_{v_i}^{ID})$, where $f_{v_i}^{ID}$ is the verb indicator (one-hot/Word2Vec [33]/Glove [40]). For new verbs, we just change the verb indicator instead of increasing MLPs. It works similar to zero-shot learning like TAFE-Net [52] and Nan *et al.* [36], but performs worse (20.86 mAP, HICO-DET Full) than the reported version (23.36 mAP).

## 4.4 Visualization

To verify the effectiveness of transformations, we use t-SNE [31] to visualize $f_u, f_h \oplus f_o, T_I^v(f_h \oplus f_o)$ for different $v$ in Fig. 4. We can find integrated $T_I^v(f_h \oplus f_o)$ obviously closer to the real union $f_u$, while the simple linear combination $f_h \oplus f_o$ cannot represent the *interaction* information. We also

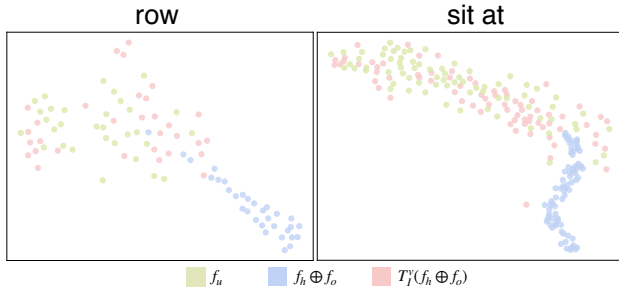
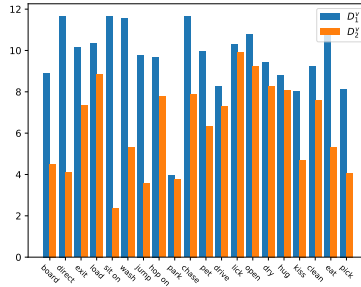

Figure 4: Visualizations of $f_u, f_h \oplus f_o, T_I^v(f_h \oplus f_o)$.    Figure 5: $D_1^v$ and $D_2^v$.

| Method | $AP_{role}^{S1}$ | $AP_{role}^{S2}$ |
|---|---|---|
| Gupta $et\ al.$ [18] | 31.8 | - |
| InteractNet [13] | 40.0 | - |
| GPNN [42] | 44.0 | - |
| iCAN [12] | 45.3 | 52.4 |
| Xu $et\ al.$ [53] | 45.9 | - |
| Wang $et\ al.$ [50] | 47.3 | - |
| TIN [27] | 47.8 | 54.2 |
| IP-Net [51] | 51.0 | - |
| VSGNet [48] | 51.8 | 57.0 |
| PMFNet [49] | 52.0 | - |
| IDN | **53.3** | **60.3** |

Table 2: Results on V-COCO [18].

| Method | Default Full | | | Known Object | | |
|---|---|---|---|---|---|---|
| | Full | Rare | Non-Rare | Full | Rare | Non-Rare |
| IDN | **23.36** | **22.47** | **23.63** | **26.43** | **25.01** | **26.85** |
| AE only | 17.27 | 14.02 | 18.24 | 20.99 | 17.39 | 22.06 |
| $T_I$ only | 21.26 | 19.96 | 21.65 | 24.73 | 23.28 | 25.17 |
| $T_D$ only | 21.05 | 19.21 | 21.60 | 24.51 | 22.56 | 25.10 |
| w/o IPT | 22.63 | 22.16 | 22.77 | 25.76 | 24.66 | 26.09 |
| w/o $L_{cls}^u$ | 19.98 | 18.02 | 20.57 | 23.24 | 20.55 | 24.04 |
| w/o $L_{cls}^{ho}$ | 21.39 | 20.08 | 21.79 | 24.65 | 22.75 | 25.22 |
| w/o $L_{bin}$ | 22.01 | 20.65 | 22.41 | 25.03 | 23.40 | 25.52 |
| w/o $L_{recon}^{AE}$ | 21.07 | 20.11 | 21.36 | 24.22 | 22.50 | 24.74 |
| w/o $L_{cls}^{AE}$ | 19.60 | 17.88 | 20.11 | 22.68 | 20.32 | 23.38 |

Table 3: Ablation studies on HICO-DET [4].

analyze the IPT. In detail, we randomly select a pair with verb $v$ and denote its features as $f_h, f_o, f_u$. Assume there are $m$ other pairs with verb $v$, whose features are $\{f_h^i, f_o^i, f_u^i\}_{i=1}^m$. Then we calculate $D_1^v = \frac{\sum_{i=1}^m \|f_u - f_u^i\|^2}{m}$ and $D_2^v = \frac{\sum_{i=1}^m (\|T_I^v(f_h \oplus f_o^i) - f_u^i\|_2 + \|T_I^v(f_h^i \oplus f_o) - f_u^i\|_2)}{2m}$. Here, $D_1^v$ is the mean distance from $f_u$ to $\{f_u^i\}_{i=1}^m$. $D_2^v$ is the mean distance from $T_I^v(f_h \oplus f_o^i)$ and $T_I^v(f_h^i \oplus f_o)$ to $f_u^i$ ($i = 1, 2, ..., m$). If IPT can effectively transform one pair to another by exchanging the human/object, there should be $D_1^v > D_2^v$. We compare $D_1^v$ and $D_2^v$ of 20 different verbs in Fig. 5. As shown, in most cases $D_1^v$ is much larger than $D_2^v$, indicating the effectiveness of IPT.

## 4.5 Ablation Study

We conduct ablation studies on HICO-DET [4] with COCO detector. The results are shown in Tab. 3. (1) **Modules**: The performance of each module is evaluated. $T_I$, $T_D$ and AE achieve 21.26, 21.05, 17.27 mAP respectively and show complementary property. (2) **Objectives**: During training, we drop one of the three validity objectives respectively. Without anyone of them, IDN shows obvious degradation, especially integration validity. (3) **Inter-Pair Transformation (IPT)**: IDN without IPT achieves 22.63 mAP, showing the importance of instance exchange policy. (4) **AE**: AE is pre-trained with: reconstruction loss $L_{recon}^{AE}$ and verb classification loss $L_{cls}^{AE}$. The removal of $L_{cls}^{AE}$ hurts the performance more severely, especially on the Rare set, while $L_{recon}^{AE}$ also plays an important auxiliary role in boosting the performance. (5) **Transformation Order**: In practice, we construct a *loop* ($f_h \oplus f_o$ to $\{f_u^{v_i}\}_{i=1}^n$ to $\{f_h^{v_i} \oplus f_o^{v_i}\}_{i=1}^n$) to train IDN with the consistency. Using $f_u$ instead of $\{f_u^{v_i}\}_{i=1}^n$, *i.e.*, $f_h \oplus f_o$ to $\{f_u^{v_i}\}_{i=1}^n$ and $f_u$ to $\{f_h^{v_i} \oplus f_o^{v_i}\}_{i=1}^n$, performs worse (**21.77** mAP).

## 5 Conclusion

In this paper, we propose a novel HOI learning paradigm named HOI Analysis, which is inspired by Harmonic Analysis. And an Integration-Decomposition Network (IDN) is introduced to implement it. With the integration and decomposition between the coherent HOI and isolated human and object, IDN can effectively learn the interaction representation in transformation function space and outperform the state-of-the-art on HOI detection with significant improvements.

## Broader Impact

In this work, we propose a novel paradigm for Human-Object Interaction detection, which would promote human activity understanding. Our work could be useful for vision applications, such as the health care system in an intelligent hospital. Current activity understanding systems are usually computationally expensive and require high computational resources, and could cost many financial and environmental resources. Considering this, we will release our code and trained models to the community, as part of efforts to alleviate the repeated training of future works.

## Acknowledgments and Disclosure of Funding

This work is supported in part by the National Key R&D Program of China, No. 2017YFA0700800, National Natural Science Foundation of China under Grants 61772332 and Shanghai Qi Zhi Institute, SHEITC (2018-RGZN-02046).

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
