[Supplementary Material]

# Supplementary for HOI Analysis: Integrating and Decomposing Human-Object Interaction

**Yong-Lu Li**[*]  **Xinpeng Liu**[*]  **Xiaoqian Wu**  **Yizhuo Li**  **Cewu Lu**[†]

Shanghai Jiao Tong University

yonglu_li@sjtu.edu.cn, xinpengliu0907@gmail.com, enlighten@sjtu.edu.cn
liyizhuo@sjtu.edu.cn, lucewu@sjtu.edu.cn

## 1  Visualized Results

Figure 1: Some HOI detection results on HICO-DET [1].

We visualize some HOI detection results of our IDN on HICO-DET [1] in Fig. 1. As shown, IDN is able to decompose and integrate various HOIs in diverse scenes and accurately detect them.

## 2  Result Analysis

We illustrate the detailed comparison between our method, Peyre *et al.* [3] and DJ-RN [2] on Rare set on HICO-DET [1] in Fig. 2. We can find that our IDN outperforms Peyre *et al.* [3] and DJ-RN [2] on various rare HOIs. The effectiveness of our IDN on Rare set proves that the dynamically learned interaction representation can greatly alleviate the data deficiency of the rare HOIs.

---

[*]The first two authors contribute equally.

[†]Cewu Lu is the corresponding author, member of Qing Yuan Research Institute and MoE Key Lab of Artificial Intelligence, AI Institute, Shanghai Jiao Tong University, China.

Figure 2: Performance comparison between our method, Peyre *et al.* [3] and DJ-RN [2] on Rare set of HICO-DET [1].

## 3  Code

We provide our source code in `https://github.com/DirtyHarryLYL/HAKE-Action-Torch/tree/IDN-(Integrating-Decomposing-Network)` under our project HAKE-Action-Torch (`https://github.com/DirtyHarryLYL/HAKE-Action-Torch`).