[Reviews · NeurIPS 2020]

Review 1

Summary and Contributions: With inspiration from brain regions integrating human and objects into HOI, the authors propose an integration-decomposition method for learning representation for interactions. Two main contributions of the method are (1) enforcing consistencies in integrating and decomposing human-object interactions (2) randomly substituting instance features as an effective way of utilizing training data. The model outperforms previous state-of-the-art methods on two commonly used benchmarks for HOI detection: HICO-DET and V-COCO.

Strengths: 1. The model utilize the consistency in integrating and decomposing HOI, which is novel in the HOI research. 2. The inter-pair transformation, or instance exchange, is an interesting way of data augmentation and could be better utilizing the HOI data. 3. The method has significantly outperformed previous state-of-the-arts on both HICO-DET and V-COCO. 4. Overall, the work should be beneficial to the HOI community.

Weaknesses: 1. Network design: In the interaction representation learning, each verb/interaction is represented by a MLP subnet T_I^{v_i} and T_D^{v_i}. Wouldn't it result in too many parameters to learn if there are a large number of verb/interaction to represent? 2. Hyperparams in v_i classification: What are the thresholds for determining if the feature distances are small or large? Do different verb classes have different thresholds? How sensitive is the final performance to these thresholds? It would be great to see such experiments and analysis. 3. Losses: My understanding of the losses in integration and decomposition validity (L_{cls}^u and L_{cls}^{ho}) is that these losses encourage the integrated and decomposed feature vectors of the ground truth verb class to be similar to f_u or f_h + f_o. How about the feature vectors of other verb classes? Shouldn't they be pushed away from f_u and f_h + f_o? The "pull and push" losses are commonly used in representation learning, while I only see "pull" here in the method. Without the "push" term, one trivial optimization outcome is that all f_u^{v_i} become mutually similar and close to f_u (so as all the f_h^{v_i} + f_u^{v_i}). How do you avoid such a trivial outcome? Is there a reason for not considering the "push" term?

Correctness: The method seems mostly correct. My concern related to the correctness of the method is the 3rd point in "Weaknesses" above. Other than that, the rest of equations look sound.

Clarity: Clear enough to understand the method and experimental analysis.

Relation to Prior Work: The paper sufficiently summarizes prior work on HOI and compared its performance with previous state-of-the-art.

Reproducibility: Yes

Additional Feedback: In terms of writing, the method part seems too long with details that can be put into supplementary materials. It would be more appreciated if more contents in the main text focus on the discussion of experiments, e.g. ablation study.


Review 2

Summary and Contributions: This paper addresses the detection of human-object interactions by proposing a novel decomposition-integration mechanism. The main idea is to model the transformation between the visual feature of separated human/object and the visual feature of the composed interaction (extracted from the tightest window enclosing both). The transformation has two directions (decomposition and integration) and is conditioned on the interaction class in both directions (Fig. 2). These transformations are realized with neural networks and can be learned with standard HOI labels. The paper also proposes a policy ("instance exchange policy") to improve the learning. Once learned, these transformations can be applied to the HOI detection problem by comparing the consistency of the decomposed/integrated features with the observed ones. Experiments are conducted on two standard benchmarks (V-COCO and HICO-DET) and the result has outperformed other SOTAs on detection accuracy.

Strengths: [+] I really like the idea of this paper as it offers a new way of thinking and addressing the HOI detection problem. Prior approaches mostly follow a fixed paradigm where you try to learn feature representation separately for human and object regions and some "interaction feature" for the interaction window, and apply a classifier to these features to predict the interaction class label. There has been many twists to this design but still most of the recent work still rely on this architecture. What this paper proposes is to not learn classifiers on the feature anymore but instead to learn transformations between individual human/object and the interaction. And finally these transformations can be applied for HOI detection. Besides its novelty, I also think the idea is interesting and sound, and is also supported by empirical evidence from the results. [+] Aside from the application to HOI detection (which seems to be a nice byproduct), the learning of these transformations itself is significant and may be the key to answer an important question: how do we model/encode interactions? Compared to traditional classifier-based approaches, this transformation mechanism seems to be a cleaner way as it nicely decouples the feature of individual human/object and the feature of interaction. [+] The proposed method has also achieved outstanding benchmark results (Tab. 1 and 2). The competition is quite intense on these dataset recently. However the method is able to outperform all SOTAs on both datasets.

Weaknesses: [-] One technical issue I had is the use of a sigmoid function to map negative distances to probabilities (L200, 226). The issue is that the range of negative distances is (-inf, 0], and therefore the domain of this sigmoid function is (-inf, 0]. The range of the sigmoid function will then be (0, 0.5], since you'll never have input value > 0. This makes sigmoid a weird choice here if you want the output to be a probability value. Seems like the exponential function is a better one since it maps (-inf, 0] to (0, 1]? *** This was addressed in the author's rebuttal (L19).

Correctness: The claims and method seem correct. The empirical methodology follows the standard protocol of the benchmarks.

Clarity: Yes, the paper is clear and easy to follow.

Relation to Prior Work: Yes, this is discussed in the related work section.

Reproducibility: Yes

Additional Feedback: * Comments: - While the proposed technique looks pretty effective on the benchmark side, I wonder if this technique (or part of this technique) can be easily applied to existing HOI detection methods and possibly achieve an immediate gain? *** This was addressed in the author's rebuttal (L21) - Following the point on the sigmoid function above, perhaps an alternative to the cross entropy loss would be a hinge loss: L=max(0, ||d^{v_i}_u - d^{v_j}_u||^2_2), where i=argmax_{v_k\in positive set} d^{v_k}_u and j=argmax_{v_k\in negative set} d^{v_k}_u? *** This was addressed in the author's rebuttal (L19) * Some prior work can be added to the SOTA comparison table (Tab. 1 and 2). This is relatively minor since adding these results does not change the standing of IDN. - Xu et al. [32]. - Tiancai Wang, Rao Muhammad Anwer, Muhammad Haris Khan, Fahad Shahbaz Khan, Yanwei Pang, Ling Shao. Deep Contextual Attention for Human-Object Interaction Detection. In ICCV, 2019. - Penghao Zhou and Mingmin Chi. Relation Parsing Neural Network for Human-Object Interaction Detection. ICCV, 2019. *** This was addressed in the author's rebuttal (L24) * Typos and errors: - L75 and Tab 1.: "Julia et al. [25]" -> "Peyre et al. [25]" - Tab. 1: "Ours" can be changed to "IDN" to be consistent with Tab. 2, 3, and 4. ##### Post-rebuttal My initial review has been quite positive and the rebuttal did not change that view. I also appreciate the authors to report extra experiments as suggested in my review in the rebuttal.


Review 3

Summary and Contributions: Updates: My major concern in the first round was 1) the paper was written verbosely. 2) I don't understand why Inter-Pair-Transformation (IPT) would lead to better HOI detection performance. For 1) I think the intro is way too complicated. The connection between the proposed approach and the Harmonic Analysis is a stretch. The author agree to simplify the intro, which would resolve my concern. For 2) Actually I doesn't understand why IPT would lead to better HOI performance? Specifically, why answering 'what is the relationship between two human-object pairs with the same HOI?' (line 20) would improve the HOI. Especially, In the ablation study, by removing IPT, the performance drops only a little. I tend to believe the author's claim that with IPT, the model could exploit shared information between different H-O pairs with the same verb. But this is based on an assumption that the verb is the most difficult to detect in the HOI. I think given the main contribution of this paper is the idea of composition and decomposition, 2) would be a minor issue. This paper tackles the task of detecting human-object-interaction from a image. The key challenge of this task is how to determine the interaction. The paper's idea is first detecting the object and human then encoding the interaction within the composition model. Therefore the paper proposes an integration-decomposition network (IDN) to learn how to combine human an object to form a pair of human object interaction. And it also learn how to decompose the human object interaction into human and object.

Strengths: Strength: 1. The composition and decomposition idea is natural to this task and interesting. The paper proposes a specific way (ineractiveness validity) to capture the interaction between human and object is novel to me. 2. The proposed method achieves good performance on V-COCO and HICO-DET.

Weaknesses: Weakness: 1. The paper is written verbosely. In the introduction, the author keep emphasizing the relation between this paper and the harmonic analysis. However, in the end, the proposed approach is just learning a composition and decomposition function. Secondly, the implementation details are floating around across the paper, which makes it pretty hard to catch the main idea. 2. After reading this paper, I still don't understand why inter-pair transformation is important for the HOI detection task? As the HOI detection task only requires the model to predict the human, object, and the interaction. Inter-pair transformation seems not relevant to the detection. 3. The proposed method, integration and decomposition network, is very similar to the paper [1, 2] in the attribute-object composition task. The related work section might need to contain a discussion to the methods in relevant tasks. [1] Nagarajan, Tushar, and Kristen Grauman. "Attributes as operators: factorizing unseen attribute-object compositions." Proceedings of the European Conference on Computer Vision (ECCV). 2018. [2] Misra, Ishan, Abhinav Gupta, and Martial Hebert. "From red wine to red tomato: Composition with context." Proceedings of the IEEE Conference on Computer Vision and Pattern Recognition. 2017.

Correctness: The method seems correct to me.

Clarity: The paper is written verbosely. Please refer to the weakenss 1.

Relation to Prior Work: The idea of the integrating and decomposing network is similar to some papers in the area of attribute-object composition. Therefore, the paper needs to discuss the difference with those paper. The paper are listed in Weakness 3.

Reproducibility: Yes

Additional Feedback:


Review 4

Summary and Contributions: The paper presents a new way of detecting human-object interaction in images. The core idea is to use 1) how well the individual object-level features can reconstruct the human-object union feature and 2) how well the union features can predict the individual human/object features as cues for predicting HOIs. The work also introduces “inter-pair transformation” for augmenting the training data. Results on two commonly used datasets (V-COCO and HICO-DET) are reported.

Strengths: **Exposition** - The analogy of HOI integration and decomposition to Fourier analysis is interesting. The method uses reconstruction errors for determining HOI. - The illustration in Figure 3 helps clarify how the method works. **Novelty** - Using feature reconstruction errors (from integration and decomposition) is an exciting and novel approach for HOI detection. **Method** - The method is technically sound. The code will be made available. **Evaluation** - The results are strong. On both HICO-DET and the V-COCO datasets, the proposed method achieves state-of-the-art performance with a sizeable boost. This highlights the effectiveness of the proposed method. - The ablation study in Table 4 evaluates essential design choices in the paper

Weaknesses: **Exposition** - I think the paper contains interesting ideas with good empirical results. However, the exposition of the method is not easy to follow and require significant revision. Here are a couple of examples that were unclear. - L6: “coherent HOI.” What does it mean to have “coherent HOI”? What are the incoherent ones? - L8: “transformations between human-object pairs.” The “transformation” is vague. Later in the paper, it turns that this is merely replacing instance-level (human or object) from similar HOI examples. The exposition is unnecessarily complicated. - The analogy between HOI analysis and Harmonic analysis is interesting at first glance, but the link is quite weak. In the problem contexts, there is only two “basis” (human and object) to form an HOI. The decomposition/integration steps introduced in this paper also do not have a close connection with the Fourier analysis as claimed. - On L33, what does the “eigen” structure of HOI mean? - On L51, “IDN can learn to represent the interaction/verb with T_I and T_D.” What does this mean? - On L205, I was not able to follow the concept of Interactive validity. There is no definition of these loss terms and no figures to illustrate this part. - Figure 2: o What does “X” mean? o g_h and g_o are not discussed. Later I found that this is just identity (swapping instance features) - Figure 3: o (a) Please specify the loss terms here. o (b) I know that the f_u^{v_i} is predicted from the concatenated feature f_h and f_o (the integration step). However, for the decomposition step, why not use f_u as input (as discussed in Eqn 1) and predict f_h and f_o? - When using the autoencoder for compressing the features f_h + f_o, isn’t that the encoded features already are “integrated”? How can we “slice” the features to get individual features? **Novelty** - The inter-pair transformation idea has been exploited in [A]. The paper should cite and discuss the differences with respect to [A] (as it was published before the NeurIPS submission). [A] Detecting Human-Object Interactions via Functional Generalization. AAAI 2020 **Method** - The proposed approach seems to require a much larger model size. For example, the method needs two (T_I and T_D) two-layer fully connected networks for *each* verb interaction. This is certainly not scalable and can be slow at test time. For example, for HICO-DET, this requires evaluating the T_I and T_D 117 times. Unfortunately, the paper did not discuss the model size and runtime performance. At least this should be discussed as a limitation. **Evaluation** - In Table 4, which dataset is this conducted on? It seems to me that this is done on the *testing set* of the HICO-DET dataset. The ablation should be done in the validation set without seeing the testing set. This may suggest that all the model tuning may also be conducted on the testing datasets, which may lead to overfitting.

Correctness: Yes, I believe so.

Clarity: I think the exposition can/should be further improved, particularly on the introduction of the method (see the Weakness section for more details).

Relation to Prior Work: The references look adequate. The paper in [A] is highly relevant to the proposed inter-pair transformation and thus should be cited. [A] Detecting Human-Object Interactions via Functional Generalization. AAAI 2020

Reproducibility: Yes

Additional Feedback: Please replace the arXiv citation with the actual venue. [9] BMVC 2018 [16] ICCV 2019 [18] CVPR 2020 [19] CVPR 2020 [21] CVPR 2020 [29] CVPR 2020 [30] ICCV 2019 [31] CVPR 2020 == Post-rebuttal comments: My main concerns in the initial reviews were 1) poor exposition and 2) scalability (model size). For writing, the authors' rebuttal agree to revise the draft accordingly to tune down the claims and clarify the method exposition. I would take leap of faith and trust the authors here. For scalability, it's clear that the method won't scale for large number of verbs as they require one MLP for *each* verb. In the rebuttal, it seems to confirm that this is actually the major source of performance improvement (e.g., using one MLP for modeling the integration-decomposition for all verbs yield > 3 mAP drop). That being said, the authors manage to have an implementation that work efficiently with a single GPU. Considering that the authors are willing to open source the code, I think this work would drive the field of human-object interaction detection forward. In sum, I would raise my rating to "above threshold".

[Author Response · NeurIPS 2020]

# Response to Submission #591 Reviews

We sincerely thank reviewers and ACs for their time and efforts. Typos will be addressed and **our code will be made publicly available**.

**To Reviewer 1**: **R1-Q1.** One MLP sub-net for one verb ($\{T_I^{v_i}(\cdot)\}_{i=1}^n$, $\{T_D^{v_i}(\cdot)\}_{i=1}^n$). **A1.** We clarify this in two aspects. (1) **Efficiency**: With carefully designed data flow, IDN can run on a **single** GPU (training: **5.0** GB, inference: **2.5** GB). We operate all transformations **in parallel** and the inference speed is **10.04** FPS (iCAN [9]: **4.90** FPS, TIN [20]: **1.95** FPS, PPDM [21]: **14.08** FPS, PMFNet [30]: **3.95** FPS). (2) **Scalability**: We also considered an implementation which utilizes **a single MLP for all verbs**, *i.e.*, conditioned MLP functions $f_u^{v_i} = T_I(f_h \oplus f_o, f_{v_i}^{ID})$ and $f_h^{v_i} \oplus f_o^{v_i} = T_D(f_u^{v_i}, f^{ID})$, where $f^{ID}$ is the verb indicator (one-hot/Word2Vec/Glove). When facing new verbs, we just change the verb indicator instead of increasing MLPs. It works similar to zero-shot learning methods like <TAFE-Net: Task-Aware Feature Embeddings for Low Shot Learning> (CVPR2019) and <Recognizing Unseen Attribute-Object Pair with Generative Model>(AAAI2019) but performs worse (**20.86** mAP on HICO-DET Full) than reported version (**23.12** mAP). We will discuss them in the final version.

**R1-Q2.** Threshold and "pull and push" losses. **A2.** (1) We directly use distances in classification (L197-200, L203-204): negative distance $-d_u^{v_i}$ acts as the **binary classification score** of $v_i$. And the gradient will make the transformation generate reasonable $\{d_u^{v_i}\}_{i=1}^n$. $-d_{ho}^{v_i}$ works similar. Thus, we did not use thresholds to measure the distances. (2) Hence, the classification losses $L_{cls}^u$ and $L_{cls}^{ho}$ can *pull* the features that meet the labels together and *push* away the others.

**R1-Q3.** Balance between method and experiment. **A3.** Thanks very much! We will revise these sections.

**To Reviewer 2**: **R2-Q1.** Exponential function and hinge loss. **A1.** Thanks! We adopt the exponential function and hinge loss, and results (HICO-DET Full) are: **Sigmoid-23.12 mAP, Exponential-23.18 mAP, Hinge-23.26 mAP**.

**R2-Q2.** IDN applied to existing HOI methods. **A2.** We apply HOI integration and decomposition to iCAN as **proxy tasks** to enhance the feature learning. The performance improves from 14.84 mAP to **18.98** mAP (HICO-DET Full). If further combining the IDN result via late fusion, it would be boosted to 23.42 mAP. Thanks for this interesting advice!

**R2-Q3.** Prior work. **A3.** We will add the results of these works to Tab. 1 and 2 in the final version.

**To Reviewer 3**: **R3-Q1.** Clarity. **A1.** We will simplify the introduction and highlight the main idea.

**R3-Q2.** The importance of Inter-Pair Transformation (IPT). **A2.** IPT is essential because it reveals the inherent nature of the implicit verb of HOI: the **shared information** between different H-O pairs with the same verb. Here, we adopt a simplified implementation, *i.e.*, human/object replacement. When replacing the human/object, transformation functions $\{T_I^{v_i}(\cdot)\}_{i=1}^n$ and $\{T_D^{v_i}(\cdot)\}_{i=1}^n$ should be **equally effective** before and after instance replacement. We can also adopt more sophisticate approaches: use motion transfer [2] to adjust the human posture according to another person with same HOIs but different posture (*e.g.*, eating while sitting/standing), change the object pose, use Wor2Vec to change the object class (similar to the method of <Detecting Human-Object Interactions via Functional Generalization, AAAI 2020>), *etc*. But these are beyond the scope of our paper. In this work, we mainly explored to leverage the integration and decomposition to learn the verb representation. We will discuss more IPT implementations in our final version.

**R3-Q3.** Attribute-as-Operator and Red-Wine. **A3.** Thanks, we will discuss them in our final version.

**To Reviewer 4**: **R4-Q1.** Method exposition. **A1.** We will revise our paper to improve clarity. The points are explained as follows. (1) *Coherent HOI* carries the interaction semantics and is **more** than the sum of isolated H and O [1], *i.e.*, the *incoherent* ones. (2) *Transformation and Analogy*: we will tune down them. (3) *Eigen*: implicit structure carrying the HOI semantics. (4) *"represent verb with $T_I$ and $T_D$"*: embed verbs in transformation model space. (5) *Interactive validity*: for $f_u$, $f_h \oplus f_o$ and $\{f_u^{v_i}\}_{i=1}^n$, we use three binary classifiers to classify them. $L_{bin}^u, L_{bin}^{ho}, L_{bin}^I$ have the same definitions ($-[ylog(p) + (1 - y)log(1 - p)]$). We will add illustration to the figure. (6) *Fig. 2*:"X" means it is hard to transform between HOI pairs directly. We will add descriptions of "X", $g_h$ and $g_o$. (7) *Fig. 3*: we will specify the losses. Eq. 1 indicates the **ideal** transformations. In practice, we construct a loop ($f_h \oplus f_o$ to $\{f_u^{v_i}\}_{i=1}^n$ to $\{f_h^{v_i} \oplus f_o^{v_i}\}_{i=1}^n$) to train IDN with the consistency. Using $f_u$ instead of $\{f_u^{v_i}\}_{i=1}^n$, *i.e.*, $f_h \oplus f_o$ to $\{f_u^{v_i}\}_{i=1}^n$ and $f_u$ to $\{f_h^{v_i} \oplus f_o^{v_i}\}_{i=1}^n$, cannot form a cycle and performs worse (**21.77** mAP on HICO-DET Full). (8) *Slice*: the encoded feature $f_h \oplus f_o$ is the **sum** of isolated H and O and thus not yet integrated (t-SNE visualization in Fig. 4). In the implementation, we did not slice $f_h \oplus f_o$ but replace H/O feature before AE compression. We will clarify this in the illustration.

**R4-Q2.** AAAI 2020 work, arXiv citation. **A2.** More please refer to **R3-Q2**. We will discuss it and fix the arXiv citation.

**R4-Q3.** Scalability. **A3.** Please refer to **R1-Q1**.

**R4-Q4.** Ablation study. **A4.** We respectfully explain that the ablation study mainly compares the performances of IDNs in different settings. In model tuning, the test set is **unseen**. Please also refer to the ablation studies of the above AAAI 2020 work, HICO-DET [3], Shen *et al.* [28], GPNN [26], iCAN [9], No-frills [16], Peyre *et al.* [25], PMFNet [30], *etc*.

[Meta-Review · NeurIPS 2020]

This paper was subject of extensive discussion post-rebuttal. In the end, all four reviewers were favor of acceptance. The AC is inclined to agree with the reviewers. The rebuttal as well as the promise of publicly available code was important for convincing the reviewers. The AC highly encourages authors to incorporate all feedback on the paper from the reviewers in their revision for camera ready since readers will have similar questions.